# Ultra-Low Thermal Conductivity of Moiré Diamanes

**DOI:** 10.3390/membranes12100925

**Published:** 2022-09-25

**Authors:** Suman Chowdhury, Victor A. Demin, Leonid A. Chernozatonskii, Alexander G. Kvashnin

**Affiliations:** 1Department of Physics, Shiv Nadar University, Gautam Buddha Nagar, Greater Noida 201314, Uttar Pradesh, India; 2Emanuel Institute of Biochemical Physics RAS, 4 Kosygin Street, 119334 Moscow, Russia; 3Skolkovo Institute of Science and Technology, Skolkovo Innovation Center, Bolshoi Blv. 30, Building 1, 121205 Moscow, Russia

**Keywords:** diamanes, Moiré structures, thermal conductivity, machine learning

## Abstract

Ultra-thin diamond membranes, diamanes, are one of the most intriguing quasi-2D films, combining unique mechanical, electronic and optical properties. At present, diamanes have been obtained from bi- or few-layer graphene in AA- and AB-stacking by full hydrogenation or fluorination. Here, we study the thermal conductivity of diamanes obtained from bi-layer graphene with twist angle θ between layers forming a Moiré pattern. The combination of DFT calculations and machine learning interatomic potentials makes it possible to perform calculations of the lattice thermal conductivity of such diamanes with twist angles θ of 13.2${}^{\circ }$, 21.8${}^{\circ }$ and 27.8${}^{\circ }$ using the solution of the phonon Boltzmann transport equation. Obtained results show that Moiré diamanes exhibit a wide variety of thermal properties depending on the twist angle, namely a sharp decrease in thermal conductivity from high for “untwisted” diamanes to ultra-low values when the twist angle tends to 30${}^{\circ }$, especially for hydrogenated Moiré diamanes. This effect is associated with high anharmonicity and scattering of phonons related to a strong symmetry breaking of the atomic structure of Moiré diamanes compared with untwisted ones.

## 1. Introduction

Miniaturization and the reduction in dimensionality of applied materials is an obvious direction towards improving the properties and efficiency of electronic devices. New materials for thermoelectric applications are among the most perspective and actively studied directions in materials science. Thermoelectric devices manipulate with heat and electricity via Seebeck, Peltier and Thomson effects, showing possible ways of future applications [1]. From this point of view, the search for perspective low-dimensional materials with required properties is an important challenge. One of the most discussed 2D materials is diamane—fully passivated bi-layer graphene with interlayer covalent bonds [2]. The atomic structure, formation mechanisms, electronic and mechanical properties of diamanes were described by Chernozatonskii et al. [3,4]. From this study, it has been revealed that hydrogenation of the outer surfaces of bi-graphene promotes the formation of interlayer bonding between the carbon atoms, making them $sp^{3}$-hybridized. It was reported that diamanes have extraordinary mechanical characteristics showing the combination of high stiffness with flexibility [3,5,6]. Additional diamanes and diamane-based structures with different types of passivation and various numbers of layers were extensively studied [7,8]. The thermal transport properties of fully hydrogenated [9] and fluorinated [10] diamanes have been studied by Zhu et al. They have observed a drastic reduction in thermal conductivity in the fluorinated diamanes compared with the hydrogenated ones. The influence of the mass of the functional group on the properties of hydrogenated and fluorinated diamanes [11] has been investigated. The structural, electronic and thermal properties of Janus diamanes in comparison with non-Janus diamanes were also theoretically studied [12], showing much lower thermal conductivity compared with non-Janus structures. The effect of mechanical stress on the phonon properties of diamane has also been found to be severe [4,13]. Special surface hydrogenation leads to the formation of diamanes with lower symmetry and having rectangular unit cells [14], which have thermal conductivity comparable to ordinary diamanes. The presence of high mechanical strength, wide electronic band gap and low thermal conductivity makes diamanes a promising material for applications as a good protective and insulating material, while diamanes with high thermal conductivity may serve as a kind of protection and heat sink for nanodevices.

It is important to note that diamanes are not hypothetical structures, they are extensively studied experimentally. Barboza et al. [15] experimentally obtained one-side passivated diamane-like material by the absorption of OH groups on the one surface of few-layer graphene. High-quality, single-layer diamonds with fluorinated surfaces were found on the CuNi(111) foil through fluorination of AB-stacked bi-layer graphene by Bakharev et al. [16]. Later, the hydrogenated diamanes were synthesized by using low-pressure and temperature process by Piazza at al. [17]. All mentioned works are devoted to study of diamanes based on AA- or AB-stacked bi-layer graphene. The variety of diamanes can be substantially expanded via consideration of bi-layer graphene, where one layer is twisted with respect to the second one on an arbitrary angle. New two-dimensional objects were proposed by Chernozatonskii et al. [18], representing a family of Moiré diamanes that have strong dependence on the physical properties on the twist angle θ [19]. These Moiré membranes possess unusual electronic band structures with ultra-wide band gaps of up to 4.5 eV, which at the same time strongly depend on the twist angle. Several theoretical [20,21] and experimental [22,23] evidences of bi-layer graphenes with different twisted angles confirm the possibility of the formation of Moiré diamanes. However, studies on the thermal transport of Moiré diamanes have not yet been carried out. We expected that the thermal properties of Moiré diamanes will strongly depend on the twist angle, which can be finely tuned during the fabrication of such structures. Moiré diamanes can be used as a thermoelectric material along with AA- and AB-stacked diamanes to build excellent thermoelectric devices. Motivated by these intriguing features of diamanes, we have studied the dependence of thermal properties of hydrogenated and fluorinated diamanes on the twist angle by using first principle calculations and machine learning interatomic potential.

## 2. Computational Methodology

### 2.1. Geometry Relaxation with Dft

The geometry optimization of considered Moiré diamanes was performed by using the VASP package [24,25,26] within the generalized gradient approximation (the Perdew–Burke–

Ernzerhof functional) [27] and the projector augmented wave method [28,29]. The plane-wave energy cutoff was set to 500 eV. The smallest allowed spacing between *k*-points was 0.25${Å}^{-1}$. The distance between the periodically located images was set to be no less than 15 Å to avoid the artificial influence of the layers on each other in a non-periodic direction. Atomic structure minimization was carried out until the change in total energy was less than $10^{-4}$ eV. The calculations of the lattice thermal conductivity were performed for fully-relaxed structures of Moiré diamanes.

### 2.2. Training of Ml Potentials via Aimd Simulations

Calculations of lattice thermal conductivity are time-consuming and are computationally very expensive, especially considering the size of the studied Moiré structures. The determination of lattice contribution to thermal conductivity requires the calculations of anharmonic third-order force constants, which requires several hundred DFT calculations. Here, the resource-consuming DFT calculations of the anharmonic force constants (third-order force constants) were replaced by calculations based on passively trained machine learning potentials.

We chose the moment tensor potentials (MTP) [30] showing an exceptionally high level of accuracy [31,32,33,34]. Calculated force constants by MTP were used in the PHONOPY [35] package to evaluate the phonon dispersion curves. Similar to classical potentials, MTP contains parameters that should be optimized during the training procedure. For each Moiré structure, we trained MTP over short ab initio molecular dynamic (AIMD) trajectories.

It was shown by Mortazavi et al. [36] that MTPs trained over short molecular dynamic trajectories can accurately reproduce the phononic properties of various 2D materials compared with DFT simulations. Following ref. [36], we prepared the training sets by conducting AIMD simulations based on the density functional theory (DFT) [37,38] within the generalized gradient approximation (the Perdew–Burke–Ernzerhof functional) [27] and the projector augmented wave method [28,29], as implemented in the VASP package [24,25,26]. The plane wave energy cutoff of 500 eV, the Methfessel–Paxton smearing [39] of electronic occupations and the Γ-centered *k*-point meshes with a resolution of $2\pi \times 0.04{Å}^{-1}$ of the Brillouin zone sampling were used, ensuring the convergence of the energy differences and stress tensors. Two configuration sets were generated by using AIMD simulation (2000 time steps) at the constant temperature of 50 K and AIMD simulations (2000 time steps) with temperature reduction from 1000 K to 200 K. The time step for simulation was chosen to be equal to 1 fs.

For more details about the training procedure and calculation of MTP forces, readers are encouraged to check ref. [40]. MTP has been very successful recently in predicting new materials and studying their lattice dynamics and thermal conductivity [36,41,42,43,44].

### 2.3. Lattice Thermal Conductivity

The lattice part of the thermal conductivity ($\kappa_{L}$) has been determined by solving the phonon Boltzmann transport equation (BTE) [45]. In this work, the BTE has been solved within the framework of relaxation time approximation of the phonon BTE. The corresponding formula for the lattice thermal conductivity is given by [46,47]:(1)$$\kappa_{L}=\frac{1}{\Omega }\sum_{iq}g_{q}v_{iq}⊗v_{iq}\tau_{iq}c_{iq}$$
where Ω is unit cell volume, $g_{q}$ is the *q*-point weight, $v_{iq}$ is the group velocity and $\tau_{iq}$ is the relaxation time of mode *i* at the point *q* of Brillouin zone.

All phonon-based calculations, including lattice thermal conductivity, were performed by utilizing a full iterative solution of the Boltzmann transport equation as implemented in ShengBTE package [46]. The fifth-nearest neighbor interactions are included to calculate lattice thermal conductivity accurately. For 2D crystals, it is known that volume is not a well-defined quantity [48]. On the other hand, the lattice thermal conductivity $\kappa_{L}$ depends on the volume of the material, which depends on the vacuum thickness along the non-periodic direction of considered 2D materials (usually along the *z*-axis). In order to properly incorporate the volume effect on $\kappa_{L}$, we need to first multiply the obtained $\kappa_{L}$ with the vacuum thickness and then divide it by an effective thickness of each structure, which we took as the distance between upper and bottom layers.

## 3. Results and Discussions

For thermal conductivity calculations, we considered hydrogenated and fluorinated Moiré diamanes based on bi-layer graphene with different twist angles θ. The structure and properties of diamanes, Dnθ, are completely determined by the structures of the parent bi-layer graphene. Twisted graphene bi-layer is described by the twist angle θ, on which the parameters of structure as translation vectors and number of atoms in unit cell are influenced [49]. The displacement of one layer with respect to another by an angle θ leads to the formation of a superlattice, which is characterized by a specific type of local atomic alignment. The same situation is observed for Moiré diamanes. The structure and properties of parent bi-layer graphene dramatically change with a change in twist angle θ and have an impact on the formation of interlayer bonds during the hydrogenation/fluorination process, after which the diamane film is formed.

Considering diamanes (Dn), one can note that the AB stacking of layers is the most dense and stable in comparison with AA-stacked structures [3]. Both AA- and AB-stacked structures have twist angles equal to zero.

The structures of twisted bi-layer graphene with 0${}^{\circ }$ < θ < ∼16${}^{\circ }$ have structural domains with AA- and AB-stacked layers. The mechanism of formation of interlayer bonds in the AA- and AB-stacked structural domains in bi-layer graphene during functionalization is similar to ordinary diamanes [3]. However, the obtained structure of Moiré diamane is not fully passivated but represents a combination of diamane domains within a bi-layer graphene matrix [50].

Twisted bi-layer graphene with ∼16${}^{\circ }$ < θ≤ 30${}^{\circ }$ has a significantly different stacking pattern. The general characteristic structural features of these bi-layer films are the same as in quasi-crystalline bi-layer grapehene with θ = 30${}^{\circ }$ [51]. Th zigzag direction of one layer almost coincides with the armchair direction of the second one, and a lot of in-plane bonds of the upper layer form “crossings” with the bonds from the bottom layer. Atoms that form these “crossings” never form interlayer covalent bonds [18,19,52].

The 2D periodic structures of twisted bi-layers from each range of θ were chosen as a base for the construction of hydrogenated and fluorinated Moiré diamanes. We considered diamanes with θ = 0${}^{\circ }$, 13.2${}^{\circ }$, 21.8${}^{\circ }$ and 27.8${}^{\circ }$, containing 2, 14, 26 and 38 carbon atoms in each unit cell layer. Twisted bi-layer graphene structures with chosen angles have the smallest unit cells among other twisted bi-layer structures [49]. Diamane with θ = 13.2${}^{\circ }$, denoted as Dn13, is a new structure that differs from the previously considered [50] consisting of small AA- and AB-stacked domains (see Appendix A). The atomic structures of the studied Moiré diamanes are shown in Figure 1, where the top and side views are shown. Black and gray atoms represent the top and bottom layers of the Moiré diamane, respectively, while orange and yellow atoms correspond to surface hydrogen/fluorine coverage of the top and bottom layers, respectively.

The thermal conductivity of any material consists of electronic and lattice contributions. Electronic contribution has a small effect on thermal conductivity at the temperature range above 300 K, so, here, we are mainly interested in the lattice part of thermal conductivity. We carried out the calculations of lattice thermal conductivity for all considered diamanes, and the dependencies of the lattice thermal conductivity values on the temperature for structures with different twist angles (27.8${}^{\circ }$, 21.8${}^{\circ }$, 13.2${}^{\circ }$ and 0${}^{\circ }$) are shown in Figure 2.

The behavior of thermal conductivity can be explained by detailed analysis of the geometry of studied films, scattering mechanisms and anharmonicity. One can see that the highest value of lattice thermal conductivity belongs to hydrogenated AB-stacked diamane ($\theta =0^{\circ }$) and is equal to 1360 W/mK). The structural reason is that Dn has an ideally ordered structure and shorter C-C bonds, leading to shorter distances for heat waves. However, the disorder of twisted diamanes causes longer bond lengths, leading to longer paths for heat wave ones, resulting in lower thermal conductivity. The unit cell is contained on only four carbon atoms and two adatoms (hydrogen or fluorine). The C-C bond length and angles between them in the AB-stacked diamane are very close to that of bulk diamond. This geometry defines the high conductivity of hydrogenated diamanes based on AA- and AB-stacked bi-layer graphenes [12]. The passivation of bi-layers by heavier fluorine atoms leads to a reduction in the lattice thermal conductivity of F-Dn to 361 W/mK in comparison with hydrogenated ones, where $\kappa_{L}=1360$ W/mK (see Figure 2a) at 300 K. the increase in θ to 13.2${}^{\circ }$ increases the disorder of the structure, leading to reduction in $\kappa_{L}$ (from 1360 W/mK to 982 W/mK at 300 K for hydrogenated diamane). The data about lattice thermal conductivity at 300 K are summarized in Figure 2b. It can be observed that $\kappa_{L}$ decreases with increasing twist angle. The structure of Dn13 consists of connected and slightly deformed AA- and AB-stacked domains (see Appendix A). The interlayer bonds are not located perpendicular to the surface of the film (in comparison with ordinary AB-stacked diamane). Thus, it has non-zero distribution of bond lengths from 1.5 Å to 1.8 Å, see Appendix A. Such disorder and distribution in the bond lengths of Dn13 leads to decreasing $\kappa_{L}$, as can be seen from Figure 2a.

The further increase in θ to 21.8${}^{\circ }$, and then further to 27.8${}^{\circ }$, increases the disorder of diamanes caused by the broader distribution of bond lengths (see Appendix A). Due to low symmetry, all carbon atoms in the unit cell of Dnθ are symmetrically non-equivalent, which is also the characteristic feature of diamane quasicrystal with θ = 30${}^{\circ }$ [52]. This leads to the most critical reduction in $\kappa_{L}$ to 35 and 32 W/mK for hydrogenated Dn21 and Dn27 structures, respectively. The decrease in thermal conductivity of F-Dnθ is not so critical. As can be seen from Figure 2a, the $\kappa_{L}$ of F-Dn21 and F-Dn27 equals 97 and 90 W/mK, respectively, which are much lower compared with AB-stacked fluorinated diamane ($\kappa_{L}$ of 360 W/mK). We suppose that heavy fluorine atoms stabilize the diamane structure, preserving high-frequency oscillations—so, the influence of structural features to thermal conductivity is lower due to the presence fluorine atoms.

An obvious way towards the reduction in thermal conductivity is increasing the phonon scattering. Wilson et al. [53] have developed a model where the twist angle was introduced as a new parameter for the description of disorder in 2D systems. It should be noted that disorder related to changes in twist angle in Moiré diamanes is responsible for the enhanced Umklapp scattering at high temperatures [54]. When the Umklapp scattering processes dominate the phonon scattering, the thermal conductivity should ideally be decreased with temperature proportional to 1/T. However, sometimes deviation can be observed from this ideal behavior, as it was observed in the case of hexagonal 2D boron nitride [55].

In order to investigate the accurate power law dependency of $\kappa_{L}\left(T\right)$, we fitted the temperature dependence of $\kappa_{L}$ shown in Figure 2a with the power law relation $\kappa_{L}∼T^{\alpha }$. The results of fitting are presented in Table 1, while the dependencies for each temperature are shown in Appendix A. One can clearly note the deviation from the ideal 1/T behavior. The values of α are −1.98 and −1.07, respectively, for hydrogenated and fluorinated Dn membranes. Hydrogenated Moiré diamanes show a lower fitting parameter α in comparison with AB-stacked diamane due to the disordered structure. As the structural disorder increases, the thermal conductivity decreases, which also affects on the temperature dependence. A similar situation in the context of thermal conductivity is observed for polymeric materials, where ordered polymeric chains have a higher thermal conductivity in comparison with disordered ones [56,57]. Moiré diamane with $\theta =13.2^{\circ }$ shows a significantly larger α due to the presence of regions with AB- and AA-stacked structures, and it can be considered as an intermediate structure between ordered and disordered Moiré diamanes, which is out of the scope of our work and is the topic of separate discussion.

An almost constant α is observed for fluorinated films with twist angles from 13.2 to 27.8${}^{\circ }$, see Table 1. The temperature dependence of the lattice thermal conductivity for fluorinated diamanes (fitting parameter α) remains almost unchanged with respect to twist angle. This can be explained by the passivation of the surface by heavy fluorine atoms that stabilize diamane structure. Thus, the influence of formed disordered carbon structures between fluorinated surfaces does not affect the behavior of lattice thermal conductivity on the temperature (fitting parameter α).

It is a well-known fact that lattice anharmonicity is inversely related to lattice thermal conductivity, which is related to different phonon scattering processes. In our calculations, we only considered the three phonon scattering processes as directly depicted in Appendix A. It can be observed that the scattering rates for hydrogenated Dn and Dn13 films are the same order of magnitude as fluorinated ones. However, for Dn27 and Dn21, a different situation is observed, where the scattering rates of hydrogenated diamanes Dn21 are more than ten times higher than those of fluorinated ones.

The scattering rates of hydrogenated Dn27 and Dn21 are much larger than corresponding rates for fluorinated Moiré diamanes (see Appendix A). As we know, the scattering rates are inversely related to the lattice thermal conductivity; the more the scattering rates, the less $\kappa_{L}$ will be. According to this relation, the lattice thermal conductivity of the Dn and Dn13 films should be larger in comparison with its fluorinated counterparts. However, this trend should be reversed for the films with twist angles of 27.8${}^{\circ }$ and 21.8${}^{\circ }$. Our discussion is confirmed by the data provided in Figure 2.

Since a fluorine atom is 19 times heavier than a hydrogen atom, the lattice part of the thermal conductivity of the fluorinated system (F-Dn) should be less than that of the hydrogenated one [10]. However, it is interesting to remark that with increasing twist angle, the hydrogenated systems become more anharmonic than the corresponding fluorinated films. As a result, the $\kappa_{L}$ values of the hydrogenated films become less than that of the fluorinated films for higher twist angles, i.e., for 27.8${}^{\circ }$ and 21.8${}^{\circ }$.

The strength of the anharmonic potential can be estimated by the frequency dependent $Gr\ddot{u}neisen$ parameter γ. The calculated $Gr\ddot{u}neisen$ parameter as a function of phonon frequency for considered Moiré diamanes is shown in Appendix A. According to these calculations, it is clearly seen that hydrogenated systems are much more anharmonic than the fluorinated ones. One can see the changes in the $Gr\ddot{u}neisen$ parameter with respect to twist angle for fluorinated and hydrogenated structures from Appendix A, where the pronounced anharmonicity of hydrogenated diamanes is clearly seen.

The variation of thermal conductivity can also be understood from the weighted phase space associated with the three phonon processes. The total phase space ($P_{3}$) corresponds to three phonon processes and can be expressed in terms of phase space corresponding to emission and absorption processes as follows:(2)$$P_{3}=\frac{2}{3\Omega }\left(P_{3}^{+}+\frac{1}{2}P_{3}^{-}\right)$$
where $P_{3}^{+}$ and $P_{3}^{-}$ correspond to absorption and emission processes, respectively. In Equation (Equation 2), Ω is a normalization factor. Here, $P_{3}^{+}$ and $P_{3}^{-}$ can be defined as: (3)$$P_{3}^{\pm }=\sum_{j}dqD_{j}^{\pm }\left(q\right)$$
where $D_{j}^{+}\left(q\right)$ and $D_{j}^{-}\left(q\right)$ are related to absorption and emission processes, respectively.

The weighted phase space of the studied twisted diamanes was calculated as shown in Appendix A. Similar to the three phonon scattering rates, here, the three-phonon scattering phase space is also inversely related to the lattice part of the thermal conductivity, leading to the fact that hydrogenated Dn and Dn13 films have lower phase-space values in comparison with corresponding fluorinated diamanes. That is why $\kappa_{L}$ is larger for hydrogenated Dn and Dn13 diamanes (Figure 2b).

On the other hand, we observed the inverse situation for Dn21 and Dn27 films: $\kappa_{L}$ is higher for fluorinated films compared with hydrogenated ones, see Figure 2b. Here, throughout the frequency spectrum, the hydrogenated systems show less phase-space scattering than the fluorinated systems (Appendix A). This implies that these fluorinated films should have higher thermal conductivity than the hydrogenated films, which is consistent with Figure 2b.

The phonon density of the states of considered diamanes are also different for hydrogenated and fluorinated ones. The calculated phonon densities of states by using moment tensor potentials (MTP) are shown in Figure 3. Presence of high-frequency modes is the feature of densities of states of all hydrogenated diamanes, see Figure 3. These modes, with a frequency of ∼85 THz, correspond to vibrations of surface hydrogen adatoms in the direction perpendicular to the surface of the film. Another important difference is related to hydrogen high-frequency vibrations that usually locate at 30–40 THz. For fluorinated systems, it is clearly seen that fluorine vibrations have lower frequencies in comparison with carbon (Figure 3). So, we can say that hydrogenated diamanes can be represented as heavy and rigid carbon cage coverings by light hydrogen atoms, while fluorinated diamanes can be represented as carbon cages covered by heavy fluorine atoms. In this case, the vibrations of hydrogen atoms on the surfaces will not influence the vibrations of the carbon frame much. From another side, the vibrational motion of fluorine atoms will be transferred to the carbon frame, leading to increased disorder in the motions of carbon atoms.

In Appendix A, we show the percentage contribution of different phonon branches to the lattice thermal conductivity. It can be clearly seen that the highest contribution to the lattice thermal conductivity comes from acoustic phonon branches, which is common for most 2D materials [58]. It should be noted that contribution from the optical branches of fluorinated Moiré diamanes monotonically decreases with increasing the twist angle (Appendix A), while hydrogenated films display the non-monotonic behavior of changes in contribution with twist angle. Moreover, there is almost no dependence of phonon contributions to lattice thermal conductivity with increases in temperature for fluorinated films. This is due to the fact that the fluorinated system is more massive in comparison with hydrogenated diamanes, as we discuss earlier.

It is known that the phonon mean free path of a particular phonon branch is directly proportional to the contribution to the lattice thermal conductivity and is directly proportional to the phonon group velocity, which inversely depends on the mass. As the fluorinated system is more massive than the hydrogenated system, that is why the group velocity is less in the fluorinated system. As a result, the phonon mean free path does not increase in the same way as the hydrogenated system does because of its lighter mass.

## 4. Conclusions

We performed a systematic investigation of thermal properties of both non-twisted diamanes (Dn) and Moiré diamanes with twist angles of 13.2${}^{\circ }$, 21.8${}^{\circ }$ and 27.8${}^{\circ }$. The calculations of lattice thermal conductivity of the studied diamanes were performed by using the combination of the ab initio method with machine learning interatomic potentials (i.e., moment tensor potentials). The strong connection of thermal properties with the geometry of Moiré diamanes was found. Thermal conductivity decreases with the increase in the disorder of structure of diamanes, i.e., increase in twist angle θ leads to an increase in disorder. Significantly lower thermal conductivity was found for hydrogenated diamanes based on bi-layers with 16${}^{\circ }$ < θ < ∼30${}^{\circ }$ than for fluorinated ones, where fluorine adatoms are responsible for the stabilization of the diamane structure. The analysis of thermal properties was performed by the calculations of the frequency-dependent $Gr\ddot{u}neisen$ parameter, three-phonon scattering rates and phase space, projected phonon densities of states and the contribution of different phonon branches to the lattice part of thermal conductivity (see Appendix A). Our analysis makes it possible to reveal that hydrogenated Moiré diamanes with twist angles of 21.8${}^{\circ }$ and 27.8${}^{\circ }$ are much more anharmonic in comparison with fluorinated ones ( both AB-stacked and Moiré), as can be seen from the larger values of the $Gr\ddot{u}neisen$ parameter at low frequencies (Appendix A). Fluorinated Moiré diamanes do not show any traces of anharmonicity according to our analysis. It is important that for the hydrogenated systems the increase in temperature leads to an increase in contribution from optical branches, while for fluorinated films such an increase in temperature is almost negligible. The thermal conductivity of Moiré diamanes drastically changes with adsorption type and twist angle of the parent bi-layer graphene and, therefore, can be easily tuned. Diamanes displaying both high and low values of thermal conductivity can be considered as prospective candidates for applications in thermal management devices [59].

## Figures and Tables

**Figure 1 membranes-12-00925-f001:**
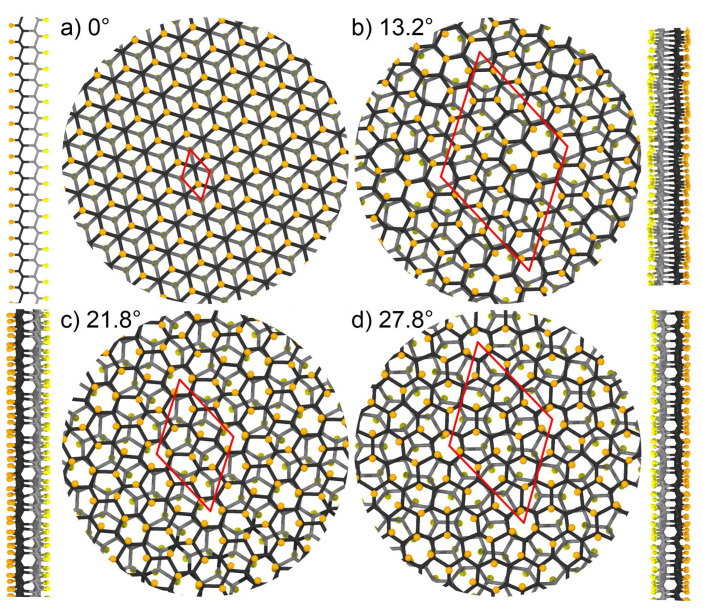
Top and side views of the atomic structure of considered diamanes with twist angles of (**a**) 0${}^{\circ }$, (**b**) 13.2${}^{\circ }$, (**c**) 21.8${}^{\circ }$ and (**d**) 27.8${}^{\circ }$. Black and orange atoms correspond to carbon and hydrogen/fluorine atoms of the top layer, while gray and yellow are carbon and hydrogen/fluorine atoms of the bottom layer, respectively. Red lines denote the considered unit cell of periodic structures.

**Figure 2 membranes-12-00925-f002:**
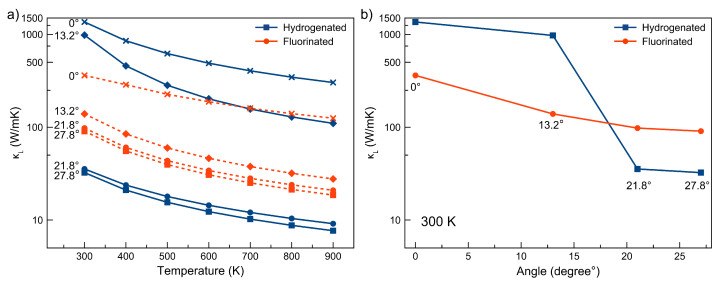
(**a**) Calculated temperature dependence of lattice thermal conductivity ($\kappa_{L}$) and (**b**) variation of $\kappa_{L}$ calculated at 300 K with twist angle of hydrogenated and fluorinated Moiré diamanes together with AB-stacked diamanes (θ = 0${}^{\circ }$).

**Figure 3 membranes-12-00925-f003:**
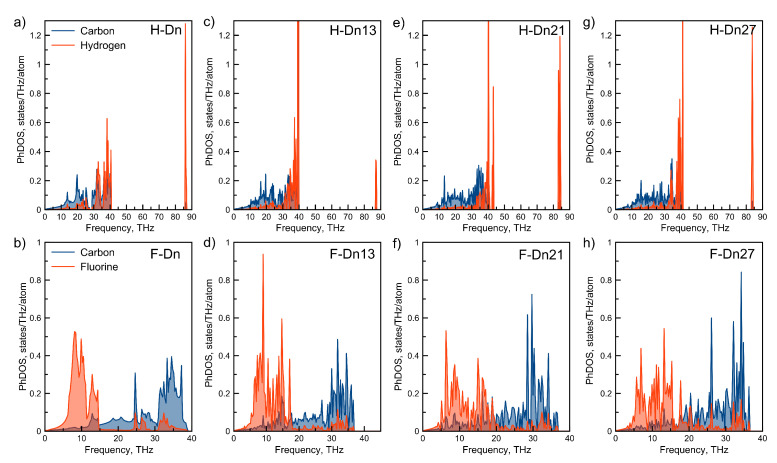
Phonon density of states projected to each atomic type calculated by using MTP for considered hydrogenated and fluorinated diamanes with twist angles of 0${}^{\circ }$, 13.2${}^{\circ }$, 21.8${}^{\circ }$ and 27.8${}^{\circ }$.

**Table 1 membranes-12-00925-t001:** Fitting parameter α of the dependence of lattice thermal conductivity on the temperature $\left(\kappa_{L}∼T^{\alpha }\right)$ for studied Moiré diamanes.

Angle θ	0${}^{\circ }$	13.2${}^{\circ }$	21.8${}^{\circ }$	27.8${}^{\circ }$
H-Dnθ	−1.98	−2.29	−1.27	−1.36
F-Dnθ	−1.07	−1.55	−1.48	−1.52

## Data Availability

Data are contained within the article and Appendix A.

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
