# Peer review of "Ultra-Low Thermal Conductivity of Moiré Diamanes"

_membranes, 2022, doi:10.3390/membranes12100925_

Round 1
Reviewer 1 Report
This is a very nice work of great interest. It complies with all the journal criteria, almost entirely. It is my understanding that the manuscript should be published after minor revision. However, some issues should be addressed:
Abstract:
Line 1: Why “quasi-2D membranes” is used to define the materials?
Line 13: “Ordinary ones” should be replaced by a more precise description.
Introduction:
Line 16: First sentence is not clear.
Line 18: First part of the sentence is not clear.
Line 48: Why the authors refer to “high-quality diamane membranes” for material found on CuNi(111) foil?
Methodology:
Line 77: Details should be given on “further calculations”.
Line 79-80: time consuming is repeated in the sentence.
Line 79-83: the sentences could be rewritten to avoid some repetition.
Typing mistakes / English: the text should be checked again. Some mistakes were detected: lines 101, 132, 134, 167, 168, 178, 180, 181, 182, 189, 194, 199, 213, 264, 266, 275, 298.
Results and discussion:
Line 160: “demonstrated” should be replaced by “shown”.
Line 180: 136 W / mK should be replaced by 1360 W / mK.
Line 194: The sentence is not clear.
Line 198-199: less should be replaced by lower.
Line 219-220: FigureS4 should be improved to support the statement given in that sentence.
Line 254-255: Figure S6 should be improved to support the statement given in that sentence.
Line 260-261: “One of the most sufficient and obvious at the same time” should be modified. The adjectives are not appropriate.
Line 274-276: The sentence should be written differently.
Conclusions: Line 298: 15 should be replaced by around 17.5.
Line 299: Careful should be removed. It should be obvious that the analysis is careful.
Line 305: See comment related to line 219-220.
Reviewer 2 Report
The manuscript by Chowdhury et al. presents interesting results on the thermal conductivity of diamanes using DFT calculations. The twist angle is varied over a range 0-27.8 degree and the thermal conductivity is calculated as a function of twist angle and discussed. The manuscript is well written and understandable. I have, however, few concerns to be addressed:
1-The heat conduction in the compounds studied occurs through chemical bonds. For the twist angle=0 degree, the path for the heat wave to be traveled (through chemical bonds) is shorter than that at higher twist angles (the worse case is the twist angle=27.8 degree). Therefore, the thermal conductivity decreases with increasing the twist angle (over the range studied). May be adding a few lines, in this respect, to the text helps the readers of the manuscript to follow the findings easier.
2-Figure 2: The temperature-dependence of lattice thermal conductivities for diamanes of one type (for example fluorinated) does not depend on the twist angle (this can be seen by comparing the curves at different twist angles). As the structure becomes less ordered (the twist angle increases), less decrease in the thermal conductivity is expected (compared to more ordered structures). This can also be understood by comparing the temperature-dependence of the thermal conductivities of solid (ordered) and liquid (disordered) phases in the literature. Similarly in the context of thermal conductivity of polymeric materials, it is known that more ordered polymeric chains have a higher thermal conductivity (this is compatible with the findings of the present manuscript) and the thermal conductivity decreases more with increasing disorder (increasing temperature); see for example Polymers 2019, 11(9), 1465; J. Chem. Phys. 135, 064703 (2011)). I recommend the authors to discuss this point in the text.
